# Design Parameters on Impingement Steam Jet Heat Transfer of Continuous Liquid Food Sterilization

**Wilasinee Sangsom and Chouw Inprasit \***

Department of Food Engineering, Faculty of Engineering at Kamphaeng Saen, Kasetsart University, Kamphaeng Saen Campus, Nakhon Pathom 73140, Thailand; wilasinee.sang@ku.th
\* Correspondence: fengchi@ku.ac.th; Tel.: +668-1803-5716

**Abstract:** The effect was clarified of the design parameters on the heat transfer of an impingement steam jet applied to continuous liquid food sterilization with the aim of high heating performance. The study investigated the effects of the steam and water Reynolds number, jet-to-target spacing to jet diameter ratio, and steam temperature on the Nusselt number. The Reynolds number was defined based on steam and water injection plate configurations in turbulent flow. The Nusselt number of the steam temperature at 120 °C was greater than at 125 °C and 130 °C and higher heat transfer was noted at a water nozzle number of two. The Nusselt number was the highest at the jet-to-target spacing to jet diameter ratio ($H/d$) of 1 and then tended to be constant for $H/d$ above 3. The present study was compared with jet impingement correlations from Huber and Viskanta, and from Martin. In addition, the Ranz and Marshall correlation of a conventional direct steam injection was compared with the impingement method. The sterilization temperature tended to increase as the steam temperature and the number of steam nozzles was increased while the number of product nozzles was decreased.

**Keywords:** steam injection; jet impingement; sterilization; heat transfer





## 1. Introduction

Continuous sterilization is the thermal processing of liquid food that is intended to destroy harmful microorganisms and undesirable enzymes. It uses high-temperature–short-time (HTST) processing to minimize product thermal damage [1–4]. Among the various heat treatments available, direct steam injection is widely used to rapidly increase the product temperature. With the direct heating method, the steam contacts directly with the product and condenses to release latent heat, resulting in a higher heat transfer coefficient. Therefore, steam injection is widely applied in the liquid food industry, such as in milk, juices, and puree [5,6]. Casoli and Coelli [6] and Shidara [7] developed the simulation of the sterilizing effect with steam injection for tomato concentrate and milk, using the Ranz and Marshall correlation.

Jet impingement is a very effective method to improve the heat transfer of thermal equipment. The effect of the jet on the surface creates a very high convection coefficient and heat transfer. When a high and uniform heat transfer rate is required over a large surface, multiple impingement jets are applied [8,9]. This system is also applied in food processing to shorten the processing time and for the extension of the product shelf life and improvement of product quality, such as in the drying of potato chips and the cooling and freezing of frozen food [10].

The flow structure of a single impingement jet has three flow regions: free jet, stagnation, and wall jet. The flow behaviors are further complicated by interactions between adjacent regions in multiple jets. The jets can be interfered with by neighboring jets before impinging on the target surface and may occur after impingement, creating an upwash flow [11,12]. The magnitude and distribution of the heat transfer coefficient depend on several parameters, including the Prandtl number ($Pr$), Reynolds number ($Re$), jet-to-jet

spacing to jet diameter ratio ($t/d$), jet-to-target spacing to jet diameter ratio ($H/d$), jet diameter ($d$), the jet geometry, the jet arrangement, and target surface [13,14].

At low $t/d$, there is significant interference between nearby jets prior to the impingement, resulting in reduced heat transfer. While $t/d$ is high, the distance between the jets is large, and this makes the impinging area wider. This results in a low area-averaged heat transfer [8,15]. The $H/d$ effect, at high $H/d$ values, produces low heat transfer because the adjacent jets interfere before reaching the impinging area [8]. In additional, the nozzle geometry, liquid flow rate, and viscosity affect the droplet size, where small droplets are the optimal spatial distribution [16].

For arrays of perforated plate impingement jets, many researchers represented the correlation of the average Nusselt number as a function of Reynolds number ($Re$) and Prandtl number ($Pr$). Here, an overview is given of the main experimental studies and correlations.

Huber and Viskanta [17] studied the effect of spent air exit in the orifice plate on the local and average heat transfer for $3 \times 3$ square array jets, at the $H/d$ = 0.25, 1.0, and 6.0 and the Reynolds number in range of 3400 to 20,500. They found that the interaction of neighboring impinged jets was reduced by spent air and the heat transfer on the target surface was greatly increased; their correlation as shown in Equation (1).

$$Nu = 0.43 Re^{0.67} Pr^{0.4} \left(\frac{H}{d}\right)^{-0.123} \left(\frac{t}{d}\right)^{-0.725} \tag{1}$$

Range of validity: 3400 < $Re$ < 20,500, $0.25 \leq H/d \leq 6$, and $4 \leq t/d \leq 8$.

Martin [18] developed the empirical equations for the prediction of heat and mass transfer coefficients. These equations were based on experimental data for single round nozzles, arrays of round nozzles, single slot nozzles, and arrays of slot nozzles. Martin's correlation presented a Nusselt function that could be used to approximate the heat transfer for arrays of round orifice plates (Equation (2)):

$$Nu = Re^{0.67} Pr^{0.42} \left[1 + \left(\frac{H}{d}\frac{F}{0.6}\right)^6\right]^{-0.05} F\frac{1 - 2.2F}{1 + 0.2(H/d - 6)F} \tag{2}$$

For plates with equally distributed pitch $t$, it followed that $F = \sqrt{\frac{\pi}{4}}\left(\frac{d}{t}\right)$.

Range of validity: 2000 < $Re$ < 100,000, $0.004 \leq F \leq 0.04$, $2 \leq H/d \leq 12$, and $1.4 \leq t/d \leq 14$.

The correlations of Huber and Viskanta and of Martin were used in the study of multiple impingement jets' heat transfer characteristics—for instance, the heat transfer of free impinging circular jets and hole channels arrays by Attalla [19], the local heat transfer distributions of confined multiple air jet impingements by Garimella [20], and the effect of circular jet array injection parameters by Yamane [15].

Usually, steam injection and steam infusion are used in the direct heating method, but, due to the high efficiency of heat and mass transfer of the impingement steam jet, the present research was interested in applying this technique for high-temperature short-time sterilization to improve product quality. The available literature on impingement jets is mostly limited to the cases of fluid injection on a target surface. Not much research has been performed on the flow characteristics of impinging jets between fluids. This paper studied the averaged Nusselt number of impingement steam jets with water and compared it to the predictive correlations of Huber and Viskanta [17] and Martin [18], who studied impingement jets in an orifice array. The experimental results were also compared to the heat transfer of conventional steam injection using the Ranz and Marshall correlation [6,7]. In addition, the water temperature and the water removal efficiency were investigated in a continuous sterilizer prototype.

## 2. Materials and Methods

### 2.1. Experimental Setup

All experiments were performed on a prototype continuous sterilizer using the impingement steam jet technique, as shown in Figure 1a, and the schematic diagram is shown in Figure 1b. The product pump (2) was used to pump the product at 60 °C from the buffer tank (1) to the impingement tank (4) to be impinged directly with the high-speed steam jets from the steam storage tank (3). Product and steam flowed through the injection plate inside the impingement tank to increase the jet speed. The product with condensate passed through the holding tube (5) for the required amount of time; then, it was immediately cool down by a vacuum cooler (6). It was kept at a suitable vacuum by a vacuum pump (7).

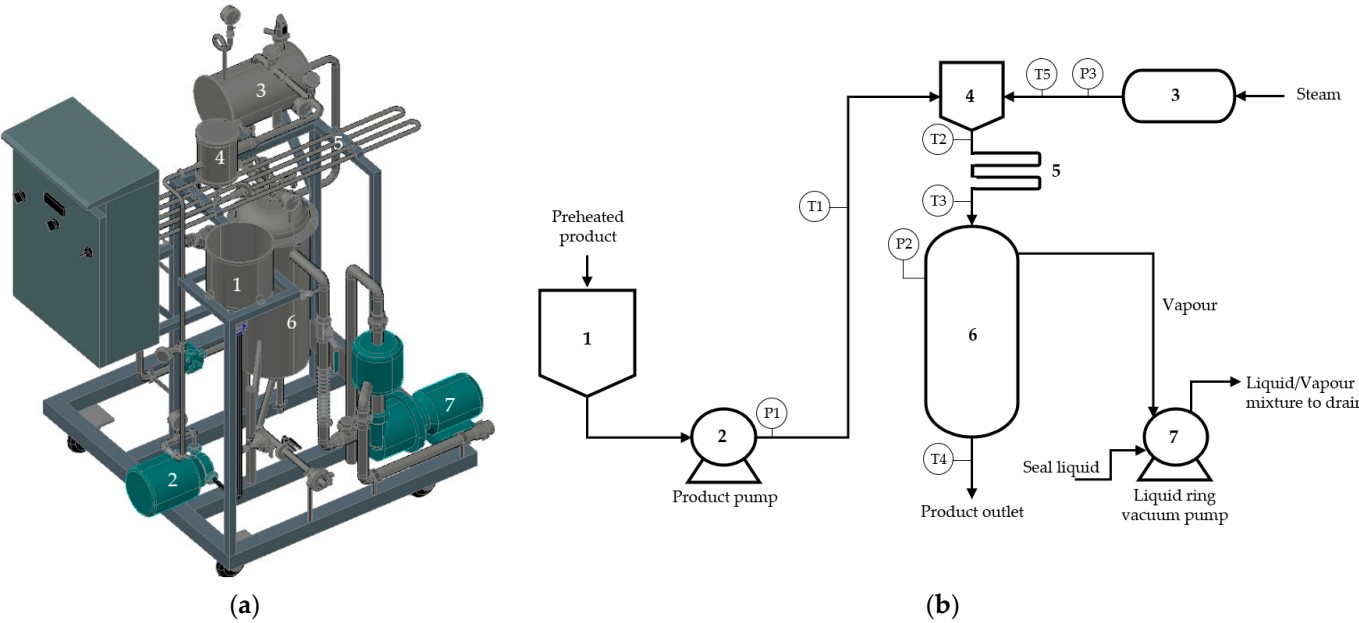

**Figure 1.** Impingement steam jet sterilizer prototype (**a**) and schematic diagram of continuous flow sterilizer using impingement steam jet technique (**b**): (1) the product buffer tank; (2) the product pump; (3) the steam storage tank; (4) the impingement tank; (5) the holding tube; (6) the vacuum chamber; (7) the vacuum pump; the temperature sensor at (T1) the product infeed, (T2) sterilization, (T3) after the holding tube, (T4) after the vacuum cooler, (T5) the steam infeed; the pressure gauges at (P1) the product pump, (P2) the vacuum cooler, (P3) the steam infeed.

The injection plates of steam and liquid food were located in the impingement tank (Figure 2a). Steam entering from the electrical boiler flowed through the injection plate with 6, 9, or 20 holes of diameter 1 mm (Figure 2b), while the product was pumped through the injection plates formed with 2, 3, or 4 holes of diameter 2 mm (Figure 2c) and the $t/d$ at 5.5 [21]. The distance of $H/d$ was tested in the range of 1–7 and was adjusted by moving the steam injection plate using the threaded socket.

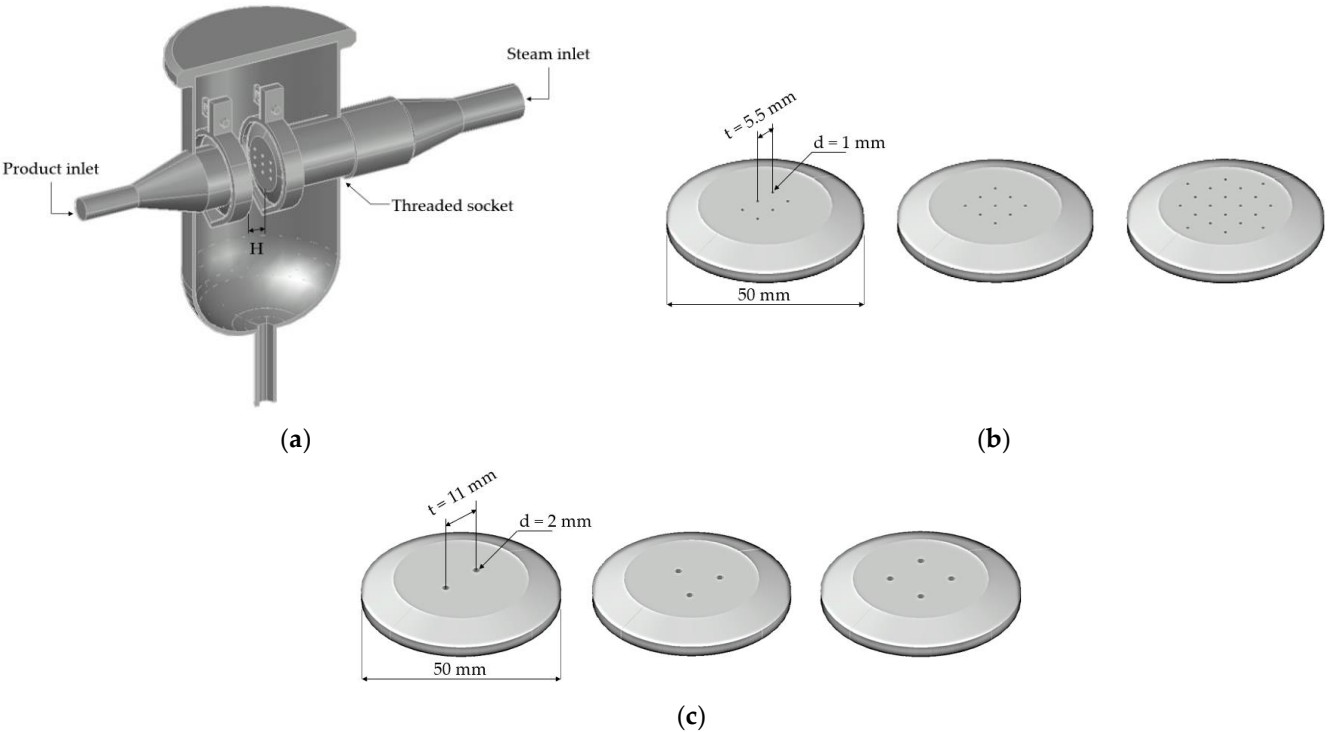

**Figure 2.** Jet impingement tank (**a**), Steam injection plates; 6, 9, and 20 holes with diameter = 1 mm (**b**), and Product injection plates; 2, 3, and 4 holes with diameter = 2 mm (**c**).

### 2.2. Measurement Procedure

In this experiment, water was used as a product to study the effect of Reynolds number (steam and product), steam temperatures, and $H/d$ values on the heat transfer in terms of Nusselt number. The results were compared with impingement correlations and steam injection correlation publications.

### 2.2.1. The Nusselt Number

The initial water temperature at 60 °C was transferred to the jet impingement tank. This was due to the assumption of a heat recovery rate of 50%, as was the case with a typical sterilization in which the cold product was pre-heated from 20 °C to 60 °C by a hot-sterilized product with indirect heating equipment [2,4]. The water flow rate was controlled at 50 kg/$h$ through the injection plates by three different numbers of nozzles at 2, 3, and 4.

The inlet steam temperatures, which were varied at 120 °C, 125 °C, and 130 °C, passed through three different numbers of steam jet nozzles (6, 9, and 20), resulting in different steam flow rates. Normally, food-grade steam or culinary steam is used for direct injection into food. However, in this experiment, the saturated steam passed through only the strainer, which typically removed visible particles in the steam.

The Reynolds numbers ($Re = d·v·\rho/\mu$) of water and steam varied in turbulent conditions ($Re > 4000$), where $d$ was the nozzle diameter, and $\rho$ and $\mu$ were the density and dynamic viscosity of water and steam, as shown in Table 1.

**Table 1.** Density and dynamic viscosity of water and steam.

| Fluid Impingement | Pressure Bar | Density ($\rho$) kg/m$^3$ | Dynamic Viscosity ($\mu$) kg/m·s |
|---|---|---|---|
| Water at 60 °C | $1.72 \pm 0.04$ | 983 | $4.66 \times 10^{-4}$ |
| Steam at 120 °C | $0.66 \pm 0.06$ | 1.22 | $1.30 \times 10^{-5}$ |
| Steam at 125 °C | $1.09 \pm 0.05$ | 1.30 | $1.31 \times 10^{-5}$ |
| Steam at 130 °C | $1.58 \pm 0.03$ | 1.50 | $1.33 \times 10^{-5}$ |

The thermal boundary condition on the target surface was that of uniform heat flux. The heat flux is determined by the thermal energy output of steam impingement, as shown in Equation (3) [22]:

$$q_w = \frac{Q}{A} = \frac{m_2 C_{p2} T_2 - m_1 C_{p1} T_1}{A} \tag{3}$$

where $q_w$ is the heat flux supplied to the target surface (W/m$^2$); $C_{p1}$, $C_{p2}$ are the specific heat values of the product infeed and product outfeed, respectively (J/kg·°C); $m_1$ is the product infeed flow rate (kg/h); $m_2$ is the product outfeed flow rate (kg/h); $T_1$ is the product temperature at the infeed (°C); $T_2$ is the product temperature at sterilization (°C); and $A$ is the target surface ($m^2$) on the product injection plate with a diameter of 59 mm.

The Nusselt number is based on the jet hole diameter ($d$), the local heat transfer coefficient ($h$, W/m$^2$·°C), the thermal conductivity of steam ($k$, W/m·°C), the heat flux supplied to the heat transfer surface ($q_w$, W/m$^2$) from Equation (3), the jet or steam temperature ($T_j$, °C), and the target surface temperature or water temperature ($Tw$, °C), as shown in Equation (4) [15]:

$$Nu = \frac{h \cdot d}{k} = \frac{q_w}{T_j - T_w} \cdot \frac{d}{k} \tag{4}$$

The study results were compared with two published impingement heat transfer correlations. The empirical correlations of Huber and Viskanta in Equation (1) and of Martin in Equation (2) have been used for the prediction of the Nusselt number; these correlations are widely used in heat transfer calculations, especially for the orifice [11]. They were correlated as functions of the steam Prandtl number ($Pr$), steam Reynolds number ($Re$), $H/d$, and $t/d$, among which $t/d$ was kept constant at 5.5 in the current study. These correlations studied steam jets impinged to the target plate, while the current research investigated steam jets impinged to turbulent water. Therefore, the current results indicated that the Nusselt numbers from both correlations were not influenced by the water Reynolds number.

The Ranz and Marshall correlation of a conventional direct steam injection was compared with the impingement method. The Nusselt number was evaluated using Equation (5) [6,7]:

$$\begin{aligned} Nu &= 2 + 0.27\, Re_p^{0.62} Pr_l^{0.33} & \left(Re_p \geq 776\right) \\ Nu &= 2 + 0.60\, Re_p^{0.5} Pr_l^{0.33} & \left(Re_p \leq 776\right) \end{aligned} \tag{5}$$

where $Re_p$ is the liquid food Reynolds number and $Pr_l$ is the Prandtl number based on the steam property.

### 2.2.2. Water Temperature

The water temperatures in the continuous sterilizer prototype at the product infeed (T1), sterilization (T2), after the holding tube (T3), and after the vacuum cooler (T4) were investigated for the varied impingement configuration and steam temperature at 120 °C, 125 °C, and 130 °C.

## 3. Results

### 3.1. The Nusselt Number

#### 3.1.1. Effect of Reynolds Number on Nusselt Number

Experiments were conducted to determine the influence of the water and steam Reynolds number on the Nusselt number at $H/d = 5$; the results are shown in Table 2. The Reynolds numbers of water injection with 2, 3, and 4 nozzles were 9476, 6317, and 4738, respectively. The Nusselt number tended to increase as the Reynolds numbers of steam and water increased because the higher Reynolds number, the more proficient the turbulence and product mixing [23,24]. In accordance with the results of Gao [25], the Nusselt numbers under $H/d = 1$ to 5 and $t/d = 4$ to 8 were significantly higher for $Re = 15,000$ compared to $Re = 5000$. Wae-hayee et al. [9] also showed that the average Nusselt number increased as the Reynolds number increased at jet Reynolds number = 5000, 7500, and 13,400 ($H/d = 2$, $t/d = 3$). The Nusselt numbers were not significantly different among water nozzles of 2, 3, and 4 at the steam injection with 20 nozzles, or the steam temperature of 120 °C, 125 °C, or 130 °C ($Re$ 18,098 to 26,009). Moreover, the water nozzle configuration of two had high heat transfer at all tested steam temperatures. Thus, the water injection of two nozzles was considered for the study of the effect of steam temperature on the Nusselt number, as shown in Figure 3.

**Table 2.** The Nusselt number at varied impingement configurations.

| Number of Steam Nozzles | Steam Temperature = 120 °C | | | Steam Temperature = 125 °C | | | Steam Temperature = 130 °C | | |
|---|---|---|---|---|---|---|---|---|---|
| | 6 | 9 | 20 | 6 | 9 | 20 | 6 | 9 | 20 |
| | $Re = 46,543$ | $Re = 34,228$ | $Re = 18,098$ | $Re = 55,721$ | $Re = 43,123$ | $Re = 20,932$ | $Re = 71,865$ | $Re = 52,984$ | $Re = 26,009$ |
| 2 water nozzles $Re = 9476$ | 183.38 ± 0.58 [c] | 129.95 ± 0.66 [b] | 59.67 ± 0.19 [a] | 190.68 ± 0.92 [b] | 139.11 ± 1.13 [b] | 63.67 ± 1.25 [a] | 204.54 ± 3.35 [b] | 146.73 ± 3.05 [b] | 67.68 ± 0.54 [a] |
| 3 water nozzles $Re = 6317$ | 175.94 ± 0.38 [a] | 135.34 ± 1.35 [c] | 60.74 ± 0.82 [a] | 183.79 ± 0.31 [a] | 139.26 ± 1.92 [b] | 63.65 ± 0.45 [a] | 199.94 ± 0.82 [a] | 139.88 ± 0.22 [a] | 68.05 ± 0.50 [a] |
| 4 water nozzles $Re = 4738$ | 179.67 ± 0.99 [b] | 123.66 ± 0.45 [a] | 60.30 ± 0.50 [a] | 190.54 ± 0.57 [b] | 130.82 ± 0.85 [a] | 63.50 ± 0.76 [a] | 200.14 ± 1.12 [a] | 141.83 ± 0.97 [a] | 68.16 ± 0.76 [a] |

Values are mean ± standard deviation, $n = 3$. [a–c] = superscript lowercase letters in columns indicate a significant ($p \leq 0.05$) difference at each the impingement condition using Duncan's new multiple range test.

The correlation of Huber and Viskanta was higher than Martin's, and the Nusselt numbers of both prediction correlations were lower than the experiment at Reynolds numbers greater than 25,000, 40,000, and 60,000 for the steam temperature at 120 °C 125 °C, and 130 °C, respectively. Moreover, the correlations from Figure 3 were formulated for the relation between the Nusselt number and Reynolds number as $Nu = c_1 Re^m$, where $c_1$ is a constant value and $m$ is the exponent of the Reynolds number at the constant of $H/d = 5$ and $t/d = 5.5$, $Pr = 1.07, 1.09, 1.10$ at the steam temperature 120 °C, 125 °C, and 130 °C, respectively; the results are shown in Table 3. The effect of steam temperature on heat transfer is clearly shown in Figure 3, while the $m$ values are shown in Table 3. The Nusselt number of the steam temperature at 120 °C was greater than at 125 °C and 130 °C because higher steam temperatures result in higher steam flow rates or steam energy input and higher heat losses from the uninsulated heating section. On the other hand, for the Huber and Viskanta correlation and Martin correlation, there was no effect of steam temperature, as shown in Figure 3.

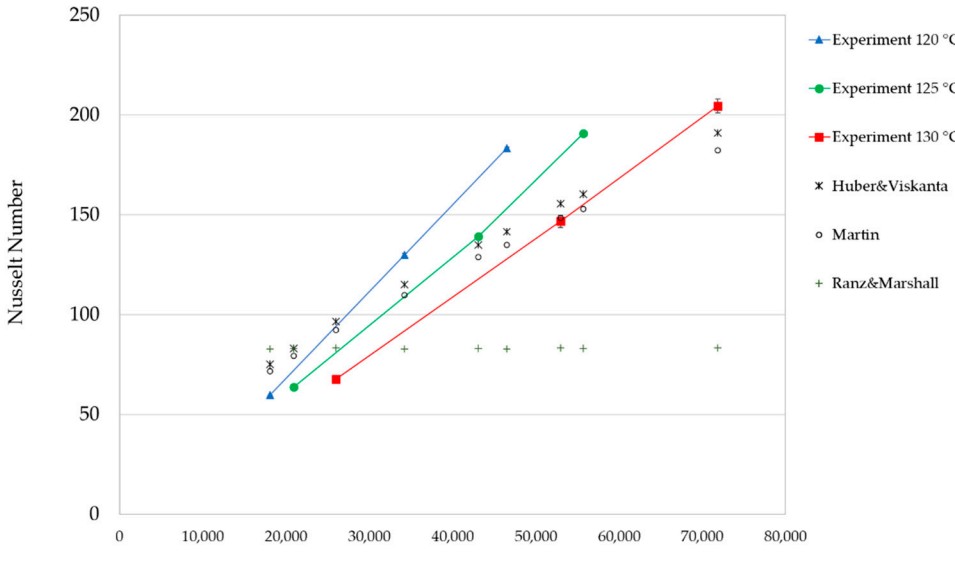

**Figure 3.** Effect of Reynolds number on Nusselt number at $H/d = 5$, water injection with 2 nozzles, and steam temperature of 120 °C, 125 °C, and 130 °C. We also show the impingement predictive correlations by Huber and Viskanta and by Martin and the conventional steam injection correlation by Ranz and Marshall.

**Table 3.** The correlations of the Nusselt number as a function of Reynolds number, $Nu = c_1 Re^m$.

| The Correlations | $c_1$ | $m$ | $R^2$ |
|---|---|---|---|
| Experiment at 120 °C | 0.0005 | 1.1936 | 0.9993 |
| Experiment at 125 °C | 0.0010 | 1.1119 | 0.9987 |
| Experiment at 130 °C | 0.0011 | 1.0880 | 1.0000 |
| Huber and Viskanta | 0.1026 | 0.6731 | 0.9999 |
| Martin | 0.9770 | 0.6733 | 0.9998 |

$R^2$ is regression statistics.

For the Ranz and Marshall correlation, the Nusselt numbers were influenced by the water Reynolds number and the steam Prandtl number according to Equation (5); thus, the Nusselt number of two water nozzles was constant at around 82.81. The Nusselt numbers of the experimental and predicted correlations of impingement steam were clearly higher than the steam injection correlation. This may be considered as strong evidence that the impingement steam jet produced high heat transfer compared with the typical steam injection method.

### 3.1.2. Effect of Jet-to-Target Distance ($H/d$) on Nusselt Number

The configuration of $2 \times 6$ water and steam jet nozzles at a steam temperature of 120 °C (the Reynolds numbers for water and steam jet were 9476 and 46,543, respectively) was investigated with different $H/d$ values. It was evident that an increase in the jet-to-target distance from an $H/d$ value of 1 to 7 led to a decrease in the Nusselt number, except for the value of $H/d = 5$, which was higher than the value of $H/d = 3$, as presented in Figure 4. Thus, the Nusselt numbers tended to be constant at $H/d > 3$. For a low $H/d$, the turbulence flow was increased, and it improved the heat transfer rate [8,15]. According to Garimella [20], $H/d = 1$ provided the highest impingement heat transfer coefficient.

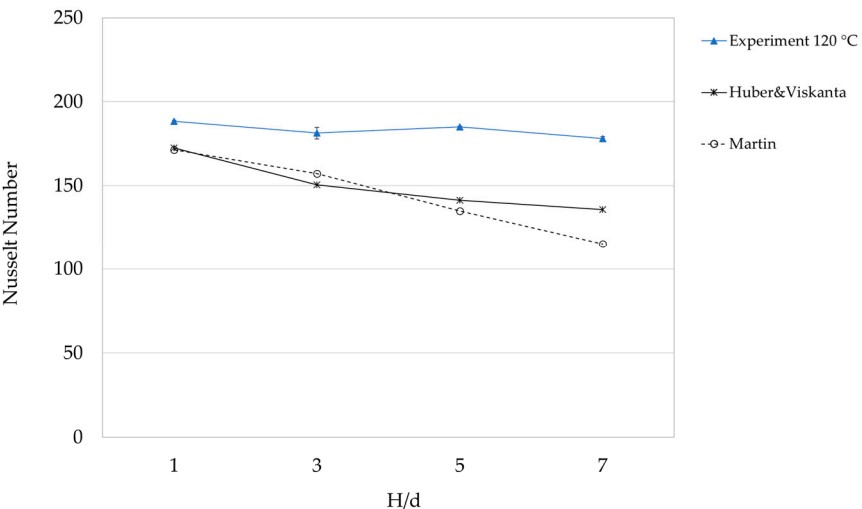

**Figure 4.** Effect of $H/d$ on Nusselt number at a steam temperature of 120 °C. We also show the impingement correlations by Huber and Viskanta and by Martin.

According to Figure 4, the relation between the Nusselt number and Reynolds number was $Nu = c_2(H/d)^n$, where $c_2$ is a constant value and $n$ is the exponent of $H/d$ at the constant of $t/d = 5.5$, $Pr = 1.07$, the steam temperature of 120 °C, and the steam Reynolds number at 46,543 (Table 4). The Nusselt numbers of both prediction correlations were lower than the number from the experiment, but Huber and Viskanta's correlation's Nusselt number slightly declined at $H/d > 3$. The trend was closer to the experimental result than Martin's correlations.

**Table 4.** The correlations of the Nusselt number as a function of $H/d$, $Nu = c_2(H/d)^n$.

| The Correlations | $c_2$ | $n$ | $R^2$ |
|---|---|---|---|
| Experiment at 120 °C | 188.03 | −0.033 | 0.6448 |
| Huber and Viskanta | 171.57 | −0.174 | 0.9952 |
| Martin | 178.13 | −0.276 | 0.8940 |

$R^2$ is regression statistics.

### 3.2. Water Temperature

The influence was explored of the impingement configuration and steam temperature on water temperatures at different positions of the process for 6, 9, and 20 steam nozzles (Figure 5a–c, respectively). The water temperatures at T2, T3, and T4 tended to increase as the steam temperature and the steam nozzle numbers were increased, in a directly proportional manner to the steam flow rate.

The effect of temperature with six steam nozzles is clearly shown in Figure 5a. The temperatures at T2, T3, and T4 for two water nozzles were higher than for three and four water nozzles, because of the increment in the Reynolds number. This was also because the mixing efficiency with the steam was increased [26]. The influence of the water nozzle number was reduced by nine steam nozzles (Figure 5b) and there was no influence with 20 steam nozzles (Figure 5c) because the steam flow rate at six steam nozzles (10.23–16.22 kg/h) was lower than with nine steam nozzles (11.29–17.94 kg/h) and 20 steam nozzles (13.27–19.57 kg/h). For this research, the temperature after the holding tube was decreased from sterilization by around 20 °C due to the prototype sterilizer having no back pressure valve to control the pressure in the holding tube. Therefore, the holding temperature was decreased from the effect of vacuum cooling.

The process of liquid food continuous sterilization involves basic stages of heating, holding, and cooling [6]. The current results indicated that the key parameters determining the heat transfer are the nozzle configuration (jet diameter, nozzle number, $H/d$), the

Reynolds number of the product and steam, and steam temperature. These parameters not only directly affected the heat transfer but also affected the water removal efficiency at the vacuum cooler. They could be used to design or scale up the sterilizer for commercial production based on the liquid food properties, production capacity, and the specified sterilizing temperature.

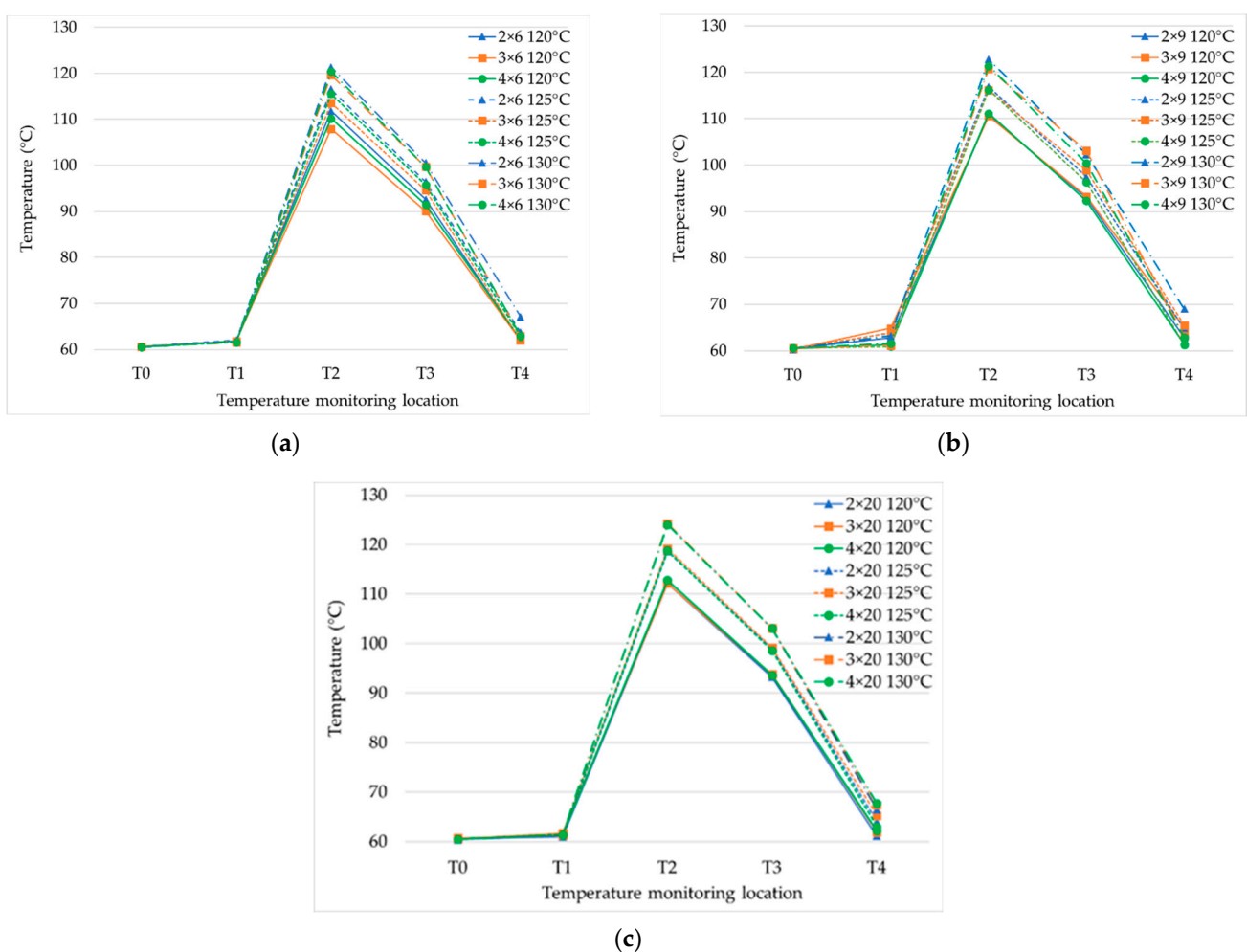

**Figure 5.** Product temperatures with various impingement configurations of jet nozzle numbers of water (2, 3, 4) and 6 steam nozzles (**a**), 9 steam nozzles (**b**), and 20 steam nozzles (**c**) at $H/d$ = 5, with steam temperature at 120 °C, 125 °C, and 130 °C.

## 4. Conclusions

This paper discussed the effects of the impingement configuration on the Nusselt number in a continuous sterilizer application. The water nozzle number of two had high heat transfer at all tested steam temperatures (Table 1), and at the steam temperature of 120 °C, the Nusselt number was the highest compared to 125 °C and 130 °C at the same steam Reynolds numbers (Figure 3). The Nusselt numbers of the Huber and Viskanta and Martin correlations were lower than the experiment at steam Reynolds numbers which were greater than 25,000, 40,000, and 60,000 for the steam temperature at 120 °C, 125 °C, and 130 °C, respectively. With the Ranz and Marshall correlation comparison, it was found that the impingement jet method produced higher heat transfer compared to conventional steam injection. For the effect of $H/d$, $H/d$ = 1 had the highest Nusselt number and then tended to be constant for $H/d$ above 3, similar to the prediction of the Huber and Viskanta correlation, where the Nusselt number was decreased slightly as $H/d$ was greater than 3. The sterilization temperature (T2) tended to increase as the steam temperature and

the number of steam nozzles were increased while the number of product nozzles was decreased. The temperatures of the product after the holding tube (T3), and after the vacuum cooler (T4), were decreased from the sterilization temperature (T2) due to the heat loss to the environment and the vacuum cooler, respectively.

The use of impingement technology led to a technological improvement in high-temperature short-time liquid food sterilization.

**Author Contributions:** Conceptualization, W.S. and C.I.; methodology, W.S. and C.I.; software, W.S.; validation, W.S. and C.I.; formal analysis, W.S. and C.I.; investigation, W.S.; resources, W.S. and C.I.; data curation, W.S.; writing—original draft preparation, W.S.; writing—review and editing, C.I.; visualization, W.S.; supervision, C.I.; project administration, C.I.; funding acquisition, C.I. All authors have read and agreed to the published version of the manuscript.

**Funding:** This research received no external funding.

**Institutional Review Board Statement:** Not applicable.

**Informed Consent Statement:** Not applicable.

**Data Availability Statement:** Not applicable.

**Acknowledgments:** This research is funded by the Graduate School Fellowship Program in Agriculture and Agro-Industry from the Agricultural Research Development Agency (Public Organization) as of fiscal year 2020 and the Faculty of Engineering at Kamphaeng Saen Scholarship for Graduate Students.

**Conflicts of Interest:** The authors declare no conflict of interest.

### Nomenclature

Parameters:

| | |
|---|---|
| $A$ | target surface (m$^2$) |
| $Cp$ | specific heat (J/kg·°C) |
| $d$ | nozzle diameter (m) |
| $h$ | local heat transfer coefficient (W/m$^2$·°C) |
| $H$ | jet-to-target distance (m) |
| $k$ | thermal conductivity (W/m·°C) |
| $m$ | product flow rate (kg/h) |
| $Nu$ | Nusselt number |
| $Pr$ | Prandtl number |
| $q_w$ | heat flux supplied to the target surface (W/m$^2$) |
| $Re$ | Reynolds number |
| $t$ | Jet-to-jet distance (m) |
| $T$ | product temperature (°C) |
| $T_j$ | jet temperature (°C) |
| $Tw$ | target surface temperature (°C) |
| $\rho$ | density (kg/m$^3$) |
| $\mu$ | dynamic viscosity (kg/m·s) |

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
