# Peer review of "Design Parameters on Impingement Steam Jet Heat Transfer of Continuous Liquid Food Sterilization"

_fluids, doi:10.3390/fluids7060185_

Round 1
Reviewer 1 Report
The authors dealt with an interesting problem that can be described from a thermodynamic point of view using mathematical relations.
The authors focused mainly on the relationships describing the fluid flow velocity using the Reynolds criteria and geometric parameters in relation to heat transfer using the Prandtl and Nusselt criteria. The results focused mainly on Re and Nu numbers and their comparison according to selected authors and experiments.
However, I did not find a comparison of the amount of heat that was transferred in the investigated process.
I did not find the results from relation (3) applied in this manuscript. I miss some output where this relationship was used. I recommend supplementing the results or not reporting it.
I recommend the authors to add Nomenclature to the manuscript.
Apparently, these are partial results of the research and I assume that the authors will continue their experiments.
At this stage, the results obtained are valuable for further research in this area.
Reviewer 2 Report
1. The novelty of the present work is not clearly described.
2. In Figure 1, schematic diagram of continuous flow sterilizer using impingement steam jet technique is not well designed. Please clarify it in the text.
3. In Equation (5), please check the formula.
4. Please double-check the data in Table 2.
5. Hiw did you obtain "The water nozzle number of 2 had a high 276 heat transfer at all tested steam temperatures and the steam temperature at 120 °C, the 277 Nusselt number was the highest compared to 125 °C and 130 °C at the same steam Reyn-278 olds numbers. "?
6. Authors can use the suggested references for better understanding the nozzle design. (https://doi.org/10.1016/j.addr.2020.10.015 ; https://doi.org/10.1038/s41598-022-10369-8)
7. Please improve the English written of the manuscript.
Round 2
Reviewer 2 Report
The manuscript is well-revised.